# Telehealth Outreach Program for Child Traumatic Stress: Strategies for Long-Term Sustainability

**DOI:** 10.3390/healthcare12212110

**Published:** 2024-10-23

**Authors:** Emily Johnson, Ryan Kruis, Rosaura Orengo-Aguayo, Rebecca Verdin, Kathryn King, Dee Ford, Regan Stewart

**Affiliations:** 1College of Nursing, Medical University of South Carolina, Charleston, SC 29425, USA; 2Center for Telehealth, Medical University of South Carolina, Charleston, SC 29425, USA; 3Department of Psychiatry and Behavioral Sciences, National Crime Victims Research and Treatment Center, Medical University of South Carolina, Charleston, SC 29425, USA; 4Department of Pediatrics, Center for Telehealth, Medical University of South Carolina, Charleston, SC 29425, USA; 5Department of Pulmonary and Critical Care Medicine, Center for Telehealth, Medical University of South Carolina, Charleston, SC 29425, USA; 6Department of Psychiatry and Behavioral Sciences, Mental Health Disparities and Diversity Division, Medical University of South Carolina, Charleston, SC 29425, USA

**Keywords:** trauma care, disparities, implementation evaluation, implementation strategy, telehealth

## Abstract

Background: There are high documented rates of exposure to traumatic events and mental health disorders among youths yet existing disparities in access to care for racial and ethnic minority youths and youths in rural communities. Trauma-Focused Cognitive Behavioral Therapy (TF-CBT) is an evidence-based behavioral health therapy for children. The delivery of TF-CBT via telehealth can decrease access to care barriers. An interdisciplinary clinical team developed a training program to guide clinicians to effectively provide TF-CBT via telehealth. The goal of this study was to describe variation in implementation processes of the telehealth TF-CBT program and identify barriers and facilitators to program implementation post-training, which were utilized to develop implementation strategies for intervention sustainability. Methods: Using a mixed-methods approach, data were collected on telehealth implementation processes and facilitators and barriers to the delivery of telehealth TF-CBT. This study was guided by an adapted implementation science framework, namely the Exploration, Preparation, Implementation, Sustainment model. Interviews and surveys were completed with clinical site leaders who had participated in the telehealth TF-CBT training. Results: Throughout clinical sites, there was varied adoption and penetration of the telehealth TF-CBT program. Facilitators to implementation included leadership and site staff buy in, community needs, and training resources, while barriers included funding, available logistical resources, and child and family involvement. Conclusions: The feedback gained from this project assisted in the development of implementation strategies for increased adoption and sustainment of TF-CBT delivered via telehealth. Strategies include ongoing interactive assistance and resource support, enhanced training for stakeholders, and program adaptations, with the goal to increase access to quality mental health care for underserved populations.

## 1. Introduction

Childhood trauma is prevalent and can have significant lifelong effects on children and their families and communities. It is estimated that 34 million children, nearly half of all children in the United States, have experienced at least one trauma in their lives, and at least 16 million have experienced more than one type of traumatic event [1,2]. Research on the relationship between adverse childhood experiences and adult health outcomes shows strong connections between stress and trauma in childhood and serious long-term behavioral and physical health outcomes, including post-traumatic stress disorder (PTSD), depression, anxiety disorders, substance use, and physical health problems [3,4,5].

Children living in rural communities are particularly vulnerable to trauma effects because of geographic isolation, poverty, and lack of access to mental health services [6]. Despite high rates of trauma exposure and subsequent mental health outcomes, many children in need of mental health care still do not receive it or do not receive evidence-based care [7,8,9]. Even greater disparities in access to care are seen across geographies (rural locations) and among racial and ethnic minority youths and those whose primary language is not English [10,11,12].

For many years, telehealth (also tele-mental health, telemedicine, and telepsychology) has offered an innovative way to address gaps in care access. Telehealth can reduce barriers to care for populations that need mental health treatment but may not typically receive it due to a lack of available providers or the costs associated with traveling to appointments [7]. In addition to distance and time barriers, children and families also report that they utilize telehealth to decrease mental health care stigma by connecting with providers via child-friendly locations such as schools and primary care centers [13]. Current research provides support that mental health services delivered via telehealth are effective in addressing numerous mental health disorders in children and adolescents, including PTSD [13,14,15,16].

Trauma-Focused Cognitive Behavioral Therapy (TF-CBT) is an evidence-based, manualized treatment protocol designed to treat post-traumatic stress and co-occurring symptoms such as depression and anxiety in trauma-exposed children aged 3–18 [17]. Over 23 randomized controlled trials across the U.S. and worldwide have documented TF-CBT’s effectiveness in treating post-traumatic stress symptoms in children and adolescents [17,18]. TF-CBT has also been shown to be feasible and effective when delivered via telehealth to trauma-exposed youths, including racial/ethnic minority youths and youths in rural locations, via published case studies and open pilot feasibility trials [19,20,21,22].

Prior studies have illustrated that while tele-mental health programs are beneficial to patient outcomes [23] and improving access to care [24], implementation can be complex and challenging [24]. Implementation science methods can identify barriers and facilitators to tele-mental health program adoption [25] and thus guide strategies to improve the widespread implementation of these programs. This is one of the first studies to utilize implementation science methods to comprehensively and systematically identify barriers and facilitators to a telehealth TF-CBT program and identify potential strategies to improve implementation outcomes.

The objective of this study was to describe variation in implementation processes of an innovative telehealth TF-CBT training program, the Telehealth Outreach Program for Traumatic Stress (TOP-TS), and identify barriers and facilitators to program implementation, which can be utilized to develop implementation strategies for program sustainability.

## 2. Materials and Methods

This study was classified as quality improvement and did not require review by the Institutional Review Board. The Consolidated Criteria for Reporting Qualitative Research (COREQ) guided methods and results reporting [26].

### 2.1. Implementation Science Framework

The Exploration, Preparation, Implementation, Sustainment (EPIS) model was utilized to guide this study] after adaptation to the TF-CBT setting [27,28]. The EPIS model includes outer and inner contextual factors and bridging and innovation factors that affect implementation processes and outcomes. In the TF-CBT telehealth setting, outer factors are external to sites delivering TF-CBT (regulatory/legal standards, community demand, and funding), while inner factors are within sites (resources, leadership, and clinician and patient characteristics) and can serve as facilitators or barriers to implementation. Innovation factors are specific to the TF-CBT program (fit and complexity), while bridging factors link inner, outer, and innovation factors (collaborative engagement of the training site and clinical sites) (Figure 1).

### 2.2. The Telehealth Outreach Program for Traumatic Stress (TOP-TS)

The TOP-TS was developed in 2015 by an interdisciplinary team of clinicians at a medical university. This program provides TF-CBT via telehealth modalities for underserved youths (racial/ethnic minorities, rural communities, and low socioeconomic status) in the United States and Puerto Rico. In addition to TF-CBT clinical services, the TOP-TS team provides education, training, and technical assistance and resources to mental health workforces delivering TF-CBT via telehealth in child advocacy centers, community mental health clinics, schools, hospitals, juvenile justice agencies, and other child-serving agencies targeting rural and underserved trauma-exposed youths and families in the US and Puerto Rico. A typical training program for instruction in the telehealth delivery of TF-CBT includes two live online sessions (average of 210 min each) completed via a live interactive webinar using videoconferencing software. Each training session includes an overview of TF-CBT treatment components, tips for telehealth treatment delivery, descriptions of additional resources available in shared online folders, and time for discussion. Follow-up consultation calls are available as an opportunity for participants to discuss challenges to TF-CBT delivery via telehealth post-training and implementation.

### 2.3. Participants

Study participants were mental health providers who attended TOP-TS live training sessions for delivering TF-CBT via telehealth within the prior two years. Participation in training within the past two years served as the cut-off for inclusion criteria to improve recall validity of participants specific to clinical experiences post-training. In addition, the TF-CBT telehealth training curriculum was developed during the explosion of telehealth services during the COVID-19 pandemic in 2020, with training sessions initiating in late 2020. Thus, when recruiting participants for the study (2022–2023), the majority of potential mental health care providers would have completed the program within the prior two years. An email invitation to complete an online self-report survey on telehealth utilization was sent to these providers. This email also asked participants if they would like to be contacted to complete a semi-structured interview, and these participants were then contacted to schedule the interview.

### 2.4. Data Collection

An 8-item quantitative and qualitative investigator-initiated self-report survey was developed to gain exploratory data on TF-CBT utilization via telehealth. Quantitative questions included the number and dates of training sessions attended, number of patients seen via telehealth in six months following training, and number of patients currently seen via telehealth. Qualitative questions were framed to identify potential facilitators and barriers to telehealth integration within sites. These open-ended questions asked clinicians to define advantageous components to launching and continuing the utilization of telehealth and provided space to report challenges to telehealth utilization in their practices. An additional question asked for the identification and description of any needed resources or support for initial and ongoing telehealth implementation [Appendix A]. The survey was pilot tested with two providers who had completed the TF-CBT telehealth training over two years prior (and thus were not eligible for the study), and verbiage was edited based on their feedback. The survey email invitation was sent to 102 mental health providers who were potential participants, and then the same survey was also sent to all interview participants after they completed an interview (if they had not previously completed the survey). The survey link opened via a webpage to a Redcap database. The 8 questions were all located on one screen, and participants scrolled down to answer all questions. Average time for completion was 8–12 min. There were no missing data as each question required a response and no identifying information was collected in the survey responses. The survey remained open from 15 July 2022 to 27 April 2023. Reminder emails for survey completion were sent four times over this time frame. All survey respondents received a USD 20 gift card for completion.

A comprehensive semi-structured interview guide with standardized questions and probes was created based on initial exploratory survey results and the identified contextual factors relevant to the telehealth TF-CBT program in the adapted EPIS model. Interview questions corresponded to the outer, inner, bridging, and innovation factors in the EPIS model with additional probing questions created based on exploratory survey results. Two post-graduate trained female interviewers (RV, MHA; EJ, PhD) not involved in the delivery of the TF-CBT program completed individual interviews via telephone in April 2023. Verbal consent from participants was provided prior to beginning interviews, and interview participants received a USD 25 gift card for participation. Interviews lasted between 21 and 32 min, and field notes were taken during and immediately following each interview. Theme saturation was sought with several methods. The interviews included a set of participants with a range of experience and utilization levels with telehealth TF-CBT methods. This allowed for comprehensive coverage of EPIS domains and a wider range of understanding of factors affecting implementation. Data collection methods during the interviews supported probing, which permitted participant reports of additional factors affecting implementation. Lastly, ongoing monitoring of field notes after each interview supported theme saturation after interview completion [29].

### 2.5. Data Analysis

Interviews were transcribed verbatim, and NVIVO software (http://www.lumivero.com/ accessed on 10 January 2023) [30] was utilized for data analysis. NVIVO software enhances organization for the qualitative coding process as it permits users to efficiently code data, classify and sort themes, and share files across platforms. A combined inductive and deductive template analysis approach was utilized [31,32] with an initial codebook developed from the adapted EPIS model [27,28]. The initial codebook included codes based on operational definitions of EPIS domains of outer, inner, bridging, and innovation factors as well as codes for implementation outcomes and strategies. Two coders independently applied the codebook to two interview transcripts and then met to discuss coding discrepancies. After discussion, two additional codes were created, and two codes were collapsed based on clarifying definitions in the EPIS model. The codebook was then utilized by the two coders independently to code the rest of the interview transcripts, with regular meetings between the coders taking place each time two additional transcripts were coded.

## 3. Results

### 3.1. Survey

Of the initial survey request, 10 surveys were completed, and 5 additional surveys were completed later by the interview participants. Thirteen participants (87%) that completed the survey reported currently seeing patients via telehealth with an average of 8 current telehealth patients (range of 2–40 current telehealth patients).

The participants’ open-ended responses identified facilitators to telehealth utilization for TF-CBT post-training, which included access to training materials, insurance reimbursement, tablets made available for patients, shorter treatment sessions, ability to screen share, flexibility for scheduling, technology support, an existing telehealth platform, and supportive leadership. The open-ended responses identified barriers to telehealth utilization for TF-CBT, which included a lack of patient technology and connectivity issues, lack of space for telehealth equipment, internet connections, summer/non-routine schedules for children, patient resistance to online modalities, and patient distraction/confidentiality concerns. Suggested resources to improve telehealth utilization included resources for teenagers, resources for children with disabilities, assistance with insurance verification, stronger internet connection, technology assistance (help with internet connection and supplies—such as loaned tablets, ear buds, and hot spots), and supply boxes to mail items to patients.

### 3.2. Interviews

Eight female mental health provider clinicians completed interviews, and of these participants, six reported actively performing telehealth visits for TF-CBT. Five participants worked as clinicians in a child advocacy center, two worked for non-profit outpatient mental health facilities, and one worked for private mental health practice. The clinicians had been working in their current roles and facilities for a range of one year to eight years.

#### Interview Results—Barriers and Facilitators to TF-CBT Implementation

Table 1 displays the barriers and facilitators to the implementation of TF-CBT via telehealth as reported by the interview participants.

### 3.3. Outer Factors

Community Resources/Access and Demand: The clinician participants described significant challenges for children living in rural areas to access crucial TF-CBT services, including a lack of available parent transportation, the inability of parents to stay for in-person therapy visits with their child due to scheduling conflicts, and lack of parent or family prioritization for therapy visits. In addition, the clinicians noted that language barriers make it difficult for some patients to receive timely in-person therapy due to wait times for available interpreters. The site clinicians described TF-CBT delivered via telehealth as a means to overcome these access barriers and meet the demand for this therapy in rural communities. However, the participants noted that offering therapy via telehealth also presents new access challenges including some families lacking stable Wi-Fi connections and electronic devices in homes to conduct telehealth visits. Additionally, background noise and other distractions were noted as an obstacle to the delivery of TF-CBT via telehealth in patients’ homes. 

Funding and Regulatory/Professional Guidelines: Regulatory policies related to funding, telehealth reimbursement, and professional practice were noted as ongoing barriers to the implementation of TF-CBT via telehealth, as there have been limitations from insurance companies on reimbursements for certain telehealth services. In addition, some clinical sites lack supplies needed for comprehensive telehealth operations, such as electronic tablets. Some clinicians reported that they had received grant funding for needed supplies, while others were considering grant proposals in the upcoming year to fund these resources. A lack of clear protocols to ensure HIPAA compliance and sound consent procedures when completing telehealth visits were also noted as ongoing complexities for telehealth visits.

### 3.4. Bridging Factors

Trainer and Clinic Partnerships: The collaborative and supportive nature of the experienced TF-CBT training team was described as vital to the telehealth training program. Trainers were noted to have a wealth of experience with TF-CBT telehealth delivery, as the participants stated that they were “insightful, knowledgeable, and top in their field” when sharing their expertise of effective care delivery across different populations. The comprehensive format of the training was also seen as beneficial as clinicians appreciated the resources provided in online shared folders, which included protocols and templates, therapy worksheets and games, videos for therapy sessions, and innovative therapy modalities. The training also provided hands-on examples of care delivery via telehealth. The participants valued the ability to contact the trainers via phone or email after the training sessions ended and valued ongoing communication and support provided as clinicians began and progressed with patient telehealth visits.

All participants agreed that the training program was an appropriate length of time, and they appreciated the condensed format which allowed for completion within a few months. The participants appreciated the format of consultation calls, which were provided as a post-training resource in which clinicians could join live online group sessions with trainers and other clinicians around the country that had recently completed the training. During the calls, the trainers provided additional updated resources for the delivery of TF-CBT via telehealth while allowing time for attendees to share experiences with patient telehealth visits. These calls allowed the clinicians to receive ongoing support and insights from the trainers and their peers.

Stakeholder Supports: In addition to the TF-CBT training team, the participants discussed other stakeholders that were key to the implementation of TF-CBT via telehealth. The steps of initiating and processing referrals needed for trauma-related counseling for children were described by varied processes, originating from child protective services, law enforcement agencies, other juvenile protective organizations, and family advocates. Family advocates work directly with families to assist with referrals for trauma care, discussions with the family to determine preferences for in-person or telehealth visits, scheduling, and identifying interpreters. School administrators and counselors were identified as key potential collaborators so clinicians could provide TF-CBT via telehealth during the school day to children, although medical and academic regulations and guidelines related to youths have been barriers to providing this care in schools. These intermediary organizations and staff were often noted as having a direct impact on volumes of children seen by the agency for TF-CBT.

### 3.5. Innovation Factors

Program Fit: The TF-CBT telehealth training program was described as an excellent fit for sites as the training met patient needs and site goals for providing TF-CBT care for rural communities. Although some participants reported that they had completed the training over a year prior, they continued to use the resources and knowledge learned and predicted continued utilization of the training content in the future.

Evidence-Based Practice Fit: Conversely, although the training was perceived as an excellent fit for site goals, the degree to which the TF-CBT methods were feasible via telehealth varied based on individual patient needs. Telehealth visits with teenage children were described as practical and very successful in providing TF-CBT as it was easier to engage older children via telehealth. However, telehealth sessions were deemed difficult to conduct with younger children and children with learning disabilities due to a lack of engagement with the screen modality and other complexities with this population. In addition, the clinicians reported a lack of ‘connection’ with patients when interacting through a screen and taking a longer time to create and build sustained rapport with their patients through telehealth. Several clinicians stated that the narrative component of TF-CBT is difficult to complete effectively via telehealth. There was also concern about children’s need for more intensive trauma counseling care beyond what can be offered virtually via telehealth.

### 3.6. Inner Factors

Resources/Technology: A primary barrier to conducting telehealth visits was the availability of technology at sites, including laptops and tablets. Another challenge was identifying a quiet clean space with a professional background to conduct telehealth visits. Otherwise, resources were not identified as barriers for clinicians to conduct telehealth visits as the comprehensive training program provided the materials needed.

Resources were shared during the training and then stored as electronic documents in online shared folders so participants could access them at later dates. These documents included checklists for determining candidates for telehealth, protocols for emergencies, confidentiality and HIPAA compliance, templates for consent forms, check-in sheets to ensure an adult was present during visits, therapy worksheets, games and videos for utilization in therapy sessions, and online resources related to the conduct of TF-CBT components. Resources were also provided in Spanish. During the training sessions, the clinicians learned other essential concepts for conducting TF-CBT via telehealth including the importance of screen sharing, guidance on performing individual components of TF-CBT, such as a trauma narrative, cultural implications, and tips for how to build rapport and ensure children stay engaged. Introducing an innovative virtual relaxation room and other relaxation and coping skills were additional beneficial skills shared.

Leadership and Clinician Attitudes: There were mixed perceptions of leadership support within clinics for offering TF-CBT via telehealth to patients. Some sites reported initial and growing support over time from clinical and executive directors to offer telehealth services, as evidenced by encouragement to complete the telehealth training and the purchase of new equipment needed to offer telehealth services. Others felt that while their supervisors would support telehealth if needed, their preference was to provide in-person therapy visits. Similarly, site clinicians also had mixed preferences for offering TF-CBT via telehealth or in person. Some clinicians favored providing TF-CBT via telehealth and were committed to providing comparable services via telehealth as they do in person. Alternatively, some clinicians preferred in-person counseling visits because of reimbursement challenges with telehealth visits and the uncertainty of potential patient responses to the telehealth format. One clinician reported that team members and leadership at her site were initially not interested in offering TF-CBT via telehealth, but after she completed training and demonstrated the resources to the team, they became supportive and receptive to offering TF-CBT via telehealth.

Child/Family Involvement: There were varied observations of parent and child involvement with participation in TF-CBT via telehealth. Higher levels of family involvement were indirectly evident when younger children were present for telehealth appointments as a family member had set up the equipment for them at home and ensured they were logged in at the correct time. Some family members were cognizant to follow up with therapists, even via text, after telehealth visits. However, similar to in-person therapy visits, low parent and family involvement was also an ongoing challenge due to competing family demands and priorities. Explaining the telehealth model to parents was important to help them understand the format and potential benefits to their child. When family members were not present at home for the telehealth visits and children were left alone, therapists reported struggles with maintaining attention and engagement with the child. The training provided tips and games to introduce to children that had low engagement via telehealth modalities (children that turned off their screens or refused to speak), hence allowing them to overcome these challenges and make progress in treatment.

Child/Family Acceptability: The interviewees reported that the majority of families and patients appreciated the ability to conduct TF-CBT visits via telehealth. While participating in telehealth entails adjustment to a novel format, parents and family members appreciated that telehealth allows for appointments to be made sooner and saves them time and gas money from driving to appointments. Although at times, it was difficult to maintain the engagement of younger children for the entire telehealth visits, this population typically liked these visits because they felt comfortable in their own rooms and spaces which, at times, helped build a personal connection between the therapist and child as the child was able to demonstrate and share their room, art work, or other valued treasures via the screen. Teenagers typically favored the telehealth format as it allowed them flexibility to be on their own schedule and help with their younger siblings at home, as needed. One therapist described that she preferred meeting with all patients in person for the first visit to evaluate their preferences for visit format and determine feasibility for the telehealth format.

#### 3.6.1. Interview Results—Implementation Outcomes and Potential Implementation Strategies for TF-CBT via Telehealth

Table 2 displays descriptions of implementation outcomes and potential implementation strategies for TF-CBT via telehealth, as reported by the interview participants.

Adoption: In terms of initial telehealth adoption, multiple clinicians described the COVID-19 pandemic as the prior impetus for incorporating virtual care into their practice. For those that had already been providing care via telehealth due to the pandemic, the TF-CBT telehealth training was described as a resource to make this care more effective, as it was noted that staff were “muddling along” with telehealth modalities prior to training. Others noted the gap in access for rural residents as an important factor influencing their initial adoption of delivering TF-CBT via telehealth.

Penetration: The extent to which telehealth had been incorporated into routine practice at each site after training varied and was dependent on individual clinic patient characteristics and also changing patient preferences. Some interviewees noted that their practice offers telehealth as an option to all referred patients, while others stated that telehealth is offered based on intake assessments and whether clinicians believe a patient can be effectively treated via telehealth. Multiple clinicians noted that the number of telehealth patients they see has decreased since the earlier days of the COVID-19 pandemic, in part due to patients’ changing preferences. Despite this varied penetration of delivering TF-CBT via telehealth, for patients that received telehealth, the clinicians noted TF-CBT as highly effective in addressing trauma, “just as successful as if it was face to face”.

Feasibility/Acceptability: The TF-CBT training program was described by all interviewees as highly beneficial regardless of the extent to which they were currently providing TF-CBT virtually. The interviewees cited potential increased access for patients as a major reason for the program’s acceptability. The interviewees regularly discussed two features of the training program they found particularly beneficial: the online resources made available after training that included protocols, checklists, and other tools and the ongoing consultation calls with other peers and training program leaders for case discussions and problem-solving strategies.

Sustainability: Finally, the site clinicians generally predicted sustaining efforts to provide TF-CBT via telehealth when appropriate. Multiple interviewees noted investments that their organizations were making to sustain telehealth efforts, including purchasing new equipment, supporting ongoing telehealth training, developing educational materials for patients and partner organizations, and pursuing funding to support the program. While generally, practices seemed interested in continuing to provide telehealth as an option, some sites noted that it was not a major priority among their organizations partly due to limited reimbursement and funds for such activity.

#### 3.6.2. Interview Results—Implementation Strategies (Table 2)

Education: When discussing the TF-CBT training program, many interviewees shared feedback on potential opportunities to enhance education and resources to further aid in the implementation of TF-CBT via telehealth. Many comments were related to the benefit and value of the expanded content provided in the online resources and training itself. Specifically, clinicians expressed a desire for support for tailoring this intervention to individual populations, including children with developmental disabilities or ADHD and those from other cultural backgrounds. Similarly, one respondent noted that the resources provided in the online folder seemed to target a younger audience and that creating more age-diverse materials might better support the use of these resources with teenagers. Others noted a desire for additional training on delivering other evidence-based training sessions via telehealth, such as problematic sexual behavior cognitive behavior therapy or eye movement desensitization and reprocessing. Finally, some expressed interest in content related to more policy and programmatic aspects of delivering TF-CBT via telehealth, such as education on models for loaning out technology to patients.

Many interviewees also mentioned interest in receiving continued support due to the ‘ever-advancing’ nature of telehealth and supportive technologies. Ideas included a refresher course, participation in peer collaboration calls, periodic ‘check-in calls’ with trainers, and regular email updates from the training team on new resources available.

Technical Assistance: For the most part, the interviewees did not note a need for ongoing technology support for the delivery of telehealth. A few discussed having information technology (IT) staff either on-site or remote who were devoted to IT needs. Others noted that most clinicians have been able to navigate IT issues that come up on their own, only escalating to IT staff when needed. One interviewee, however, noted that more help with technology aspects of telehealth, including sharing screens and accessing online resources, would be beneficial.

## 4. Discussion

Childhood trauma, specifically in underserved populations, is prevalent and associated with long-term detrimental outcomes [1,2,6]. There are documented disparities in access to mental health care [10,11,12], and the delivery of mental health care services via telehealth, specifically TF-CBT, has demonstrated improved access to care and patient outcomes in vulnerable populations [16,19,20,22]. However, there are challenges related to the implementation of tele-mental health care [22,24,33]. An innovative TF-CBT training program focused on telehealth delivery was developed to overcome these challenges and improve the implementation of this modality. This study utilized implementation science principles to explore the implementation processes and outcomes of TF-CBT delivery via telehealth in a variety of settings after clinicians’ completion of the training program.

This study identified that a key facilitator of the successful implementation of TF-CBT via telehealth in clinics was the availability of applicable resources and support provided by the training team. Participants benefited from education provided by the training team during live training sessions and ongoing consultation calls, as well as online folders of resources that could be utilized in future clinical patient visits via telehealth. This valuable type of resource support is consistent with a report by Conradi et al., which described a process of rapidly transitioning mental health services from the in-person format to telehealth during the COVID-19 pandemic [33]. In this environment, site leaders facilitated multi-faceted training which included education on conducting practice initiatives via telehealth and leveraging resources such as protocol documents and checklists via online folders, contributing to an overall smoother transition process for this clinic [33]. Similarly, Appleton et al. found that ongoing support through technical assistance or ongoing consultation tended to be strongly aligned with the successful implementation of tele-mental health [34]. Related to this concept is the benefit of leadership support to facilitate tele-mental health services. The advantage of supportive leadership was evident in our study through site leader encouragement for clinicians to participate in TF-CBT training and through leaders’ provision of technology and supplies needed for transitions to the telehealth format; the benefits of leadership support for the implementation of tele-mental health has also been discussed in prior studies in the literature [21,33]. The presence of strong leadership has been demonstrated to improve the adoption of evidence-based practices in general mental health services [35,36], thus emphasizing the benefit of leadership support in the implementation of tele-mental health.

One of the primary barriers to the implementation of tele-mental health services in our study was a lack of availability of technology and internet connectivity issues at patients’ homes. Patient engagement was also discussed as a challenge by multiple participants as it was difficult to interact and engage with patients via a telehealth screen; it was concluded that some patients had needs that may be more favorably served with in-person treatment services. These barriers of technology and connectivity in patient homes, as well as the assessment that the delivery of mental health services via telehealth may be ideal for specific populations, are consistent with previous reports from the literature [24,33,37], some of which identified that young children with attentional difficulties and risk concerns [24,33] may be treated more effectively with in-person therapy.

Despite the implementation barriers to telehealth TF-CBT that were noted in our study, the participants had favorable impressions of the delivery and long-term sustainability of TF-CBT via telehealth as it meets patient demand and can increase access to quality mental health care for vulnerable populations. Thus, strategies for telehealth TF-CBT can be created to enhance the implementation of this critical innovation [38,39,40].

Based on feedback from the clinician participants in our study, ongoing training and education would be valuable resources for the implementation and sustainment of TF-CBT via telehealth. This long-term education and support of stakeholders could include ongoing scheduled consultation calls, strategically planned every six months for all training participants, and continued leveraging of resources. The calls could include updates on new resources and protocols for clinical practice and technology and offer continued collaboration with other providers to share clinical experiences. The online folder of resources can continually be updated with new clinical protocols and resources to meet dynamic health care needs, as well as strategies for identifying ideal formats for mental health service delivery for potential patients when deciding between in-person versus telehealth modalities. Importantly, the training program would ensure all stakeholders receive timely communication (emails, shared folder notifications, etc.) for announcements of newly scheduled calls and training sessions and updated available materials.

As the importance and benefit of leadership in the implementation of new tele-mental health innovations is evident, the training program can include facilitation on developing stakeholder interrelationships with focus on the identification of a leader at clinical sites who will champion and support the delivery of TF-CBT via telehealth. This process can include continued focused contact with site leaders to highlight the value of TF-CBT via telehealth in meeting the health priorities of their clinic and the population being served. Continued communication and involvement with TF-CBT telehealth intermediary contacts such as referring organizations and family advocates is also key to continued stakeholder engagement.

The training program can also be adapted and tailored to a variety of patient needs and contexts to maximize effectiveness for diverse populations. Due to the sensitive nature of TF-CBT, adaptability is a vital component of TF-CBT delivery via telehealth. Some clinician participants requested additional resources for utilization with older children (age-appropriate for teenagers) and children with development disabilities, as well as cultural implications related to TF-CBT delivery. By continuing to adapt and tailor resources, the training program can facilitate the expansion of TF-CBT via telehealth to a broader and more diverse population.

Addressing the barriers to the delivery of TF-CBT via telehealth presented by limited connectivity or a lack of technology in patient homes should be addressed; of interest is that technology and connectivity barriers have increasingly become a focus among programmatic and policy leaders [41,42]. The federal government has invested funding to enhance access to high-speed internet and has multiple initiatives targeting individual patient access to internet and technology, such as the Affordable Connectivity and Lifeline programs [43,44]. Moreover, many philanthropic and industry-led initiatives have adopted digital health equity as a funding priority, creating partnerships and grantmaking opportunities to enhance access to digital health and other services among local communities [45,46]. Leveraging free internet at libraries, community centers, and schools is another emerging practice to support enhanced telehealth access [42,47]. Incorporating information on these programs, funding opportunities, and promising practices in TF-CBT telehealth training and online resources may better equip providers to help their patients navigate the digital divide.

Lastly, the training program can highlight methods to engage patients in participation with TF-CBT by demonstrating the value and convenience of this treatment for families. Patient perceptions of the telehealth format were discussed as a barrier to implementation, similar to prior studies that documented patient resistance to online modalities for the delivery of medical treatment [48]. Innovative direct marketing strategies to deliver the purpose and benefits of this program to patients can help alleviate this obstacle to telehealth modalities. In addition, resources focused on engaging young children during the telehealth visit would be advantageous to clinicians to overcome distractions experienced with telehealth in this population, especially considering that engaging young children in tele-mental health is often perceived among clinicians to be a barrier to care [49].

This study had a few limitations as it did not include parents or children to understand their direct perceptions on the delivery of TF-CBT via telehealth. In addition, the timing and relevance of this training was influenced by the COVID-19 pandemic. While many telehealth payment flexibilities have become permanent, there is still uncertainty regarding the long-term role of telehealth in the provision of health care. Thus, perceptions regarding its utility and challenges in implementation are likely to evolve as polices are further solidified. Lastly, five of eight interview participants worked within child advocacy centers, which may limit the generalizability of the results. However, the other three interview participants worked in different environments, and all eight participants expressed varied ranges of time periods for their current work roles and varied utilization of TF-CBT via telehealth, thus demonstrating diverse experiences and perceptions. Lastly, despite the survey remaining open for 9 months (July 2022–April 2023) and four reminder emails being sent during this time frame, there was a low response rate, most likely due to hectic and busy clinician schedules. Despite the limitations, this study utilized implementation science principles to describe moderators to the implementation of TF-CBT via telehealth after the completion of the training program, which were used for the identification of implementation strategies to maximize the implementation and sustainment of TF-CBT via telehealth. Future research can include larger sample sizes to increase the generalizability of the results as well as studies to test the effectiveness of strategies to improve intervention outcomes.

## 5. Conclusions

TF-CBT via telehealth can decrease access to care barriers for trauma-affected vulnerable populations and improve patient outcomes. The COVID-19 pandemic created an immediate need to launch telehealth programs to protect the public. However, many mental health clinics lacked the necessary training, tools, and support to implement a telehealth program. Our implementation study showed that leadership and staff buy in, community need, offering practical training resources, live training sessions, ongoing consultation, and funding to assist with access to technology (tablets and internet) were key facilitators in the successful implementation of TF-CBT via telehealth. A lack of access to technology and privacy, reimbursement challenges, and difficulties with engaging younger children or those with developmental disabilities were key barriers. Providing training, technical assistance, and tailored support in the delivery of TF-CBT via telehealth is a promising implementation strategy that can help increase access to quality mental health care for underserved populations.

## Figures and Tables

**Figure 1 healthcare-12-02110-f001:**
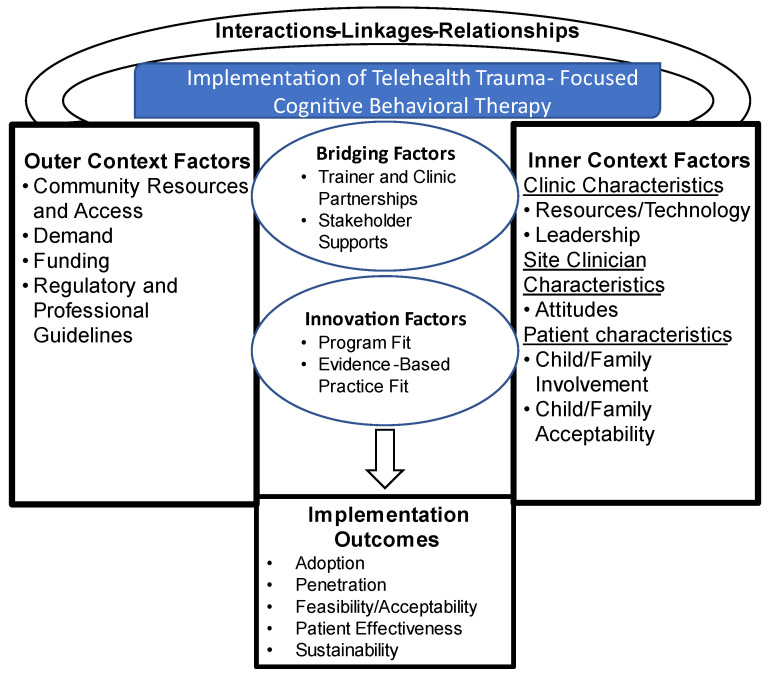
Framework for implementation of telehealth Trauma-Focused Cognitive Behavioral Therapy.

**Table 1 healthcare-12-02110-t001:** Exemplary quotations: barriers and facilitators to TF-CBT implementation via telehealth.

**External Factors**		
**Code**	**Definition**	**Quotations**
Community Resources/Access	Patient and family barriers to accessing physician offices (for in-person counseling visits) and resources needed for telehealth visits	“I think because of our population…it is a rural area and it’s very spread out, so we have a lot of transportation barriers…If they did have transportation, getting gas money to come to sessions, that was a barrier.” (Clinician 7) “We get to access communities we couldn’t access if we didn’t have telehealth, like the Spanish-speaking community. We don’t find a lot of bilingual clinicians across the state, so just having ability to connect with them via telehealth…breaks that barrier of them not getting services.” (Clinician 2) “I do face one challenge…is their access to just the technology portion, access to devices where they can connect to telehealth services. And of course, everyone has a phone, but not everyone has access to internet from home.” (Clinician 2) “So sometimes the connections work…they don’t always work well, or just noise or distraction…and trying to get them into a quiet place.” (Clinician 1)
Regulatory and Professional Guidelines	Policies related to telehealth, including reimbursement policies	“So our site, with our funding and so forth, preferred to have in person, and just with reimbursements. If it has to be telehealth, it has to be a video call on a certain platform. So they prefer in person, but sometimes there is a reason or an ability to do telehealth.” (Clinician 1)
**Bridging Factors**		
Trainer and Clinic Partnerships	Operational style telehealth training team, strategies for support provision, and involvement of clinicians in training and implementation processes	“The trainers were excellent…very knowledgeable. They were top in their field, and really had a lot of good insights, even clinically some tools of what to do, how to help kids, how to get them to open up…how to continue to focus on the TF-CBT. They were very skilled and very helpful.” (Clinician 1) “I liked that I could always access [trainers], especially once I start seeing clients, and just for their knowledge base within the systems already.” (Clinician 4) “… just their experience…They’ve been doing it a lot…have great experience with different things that worked and didn’t work and what pitfalls can happen…” (Clinician 6) “I also like that they went through a trauma narrative…we’re going to write it together, so they can see it as it’s being written, just like they would in the office. So really trying to mirror an office experience to what they’re getting in telehealth.” (Clinician 4) “They were always happy and willing for us to email them and ask questions. The consultation calls were really helpful, so we could problem solve together and talk about cases that we were having issues with…” (Clinician 5) “The [consultation calls] were helpful, because I was able to connect with other clinicians, and in a sense offer me some normalizing, that a lot of my challenges were also faced by other clinicians. And then sharing resources, sharing ways of doing things among each other, and obtaining guidance from the trainers on how to implement certain services.” (Clinician 2) “I think it was good. And then having the length of time was really good. And then having the calls afterwards so if we had questions and being able to build on what we have learned over the course of them working through the practice skills of TF-CBT. And providing and showing different resources that were available to us, maybe that we hadn’t had time to play around with, was very helpful. And them being able to bounce ideas off of each other about what was working for us, what wasn’t working for us, and getting feedback from our peers.” (Clinician 3)
**Innovation Factors**		
Program Fit	Degree to which telehealth training and TF- CBT program (content and complexities) match and fit goals, values, and tasks of clinic and staff	“It was very clear, many resources, many ideas, ways to interact with people via telehealth, and many of those were very practical and useful with in person therapy as well.” (Clinician 1) “I think it’s still going to continue to be needed, be used. And it helps many people that live far, especially in rural areas, or even city areas, where transportation and time is different. Our culture is people want things quick, fast, close, convenient kind of thing, and it fits that.” (Clinician 1) “I thought it was great, it followed our practice of care. It gave us an outline for safety, like for those steps in creating what needed to happen to create the [program]; how we can get in touch with people and what was needed and required to provide the services.” (Clinician 6)
Evidence-Based Practice Fit	Degree to which TF-CBT practices are feasible in site settings and perceptions about advantages and complexity of child trauma care delivered over telehealth	“With older kiddos, I had all sorts of great success, definitely…the treatment we provide has been just as successful as if it was face to face. I have families I’ve been able to engage very well through telehealth, and I have never met them in person. It’s all been through video calls.” (Clinician 2) “…doesn’t necessarily work with the little ones. We have tried unsuccessfully. So I’m going to say in the five to seven age range, it’s really hard for a child to connect fully on a video call. So that has been a barrier for me…So I’d rather see those children in person.” (Clinician 2) “So far I’ve not seen any younger kiddos. But…I’m a people person as I like to have a person in front of me. I’ve struggled with initial building rapport…it feels a little less personal. I don’t know if it’s just me or if the kiddo’s…maybe picked it up from me, or if they feel it as well…” (Clinician 3) “…I think that was and is my biggest concern with dealing with trauma with kids. I want to make sure that the rapport is there before we move into the heavy stuff. And I think it’s harder through telehealth to build that rapport.” (Clinician 3) “No, not totally successful. I think there have been some gains for my clients, but for the most part that connection is missing. And the ability to build trust that you need to be able to do, like the trauma narrative, is less than it would be if it was in person…” (Clinician 8) “But the connection piece I think, is the biggest piece. I think that it’s easier to do things together when you’re in the room together…” (Clinician 8) “In case for some reason they are needing additional support than what I can give them virtually…” (Clinician 4)
**Inner Factors**		
Resources/Technology	Level of resources and technology dedicated to adoption and ongoing telehealth operations; includes resources provided by training team	“I’m hoping as soon as we can get technology secured that I can go ahead and move forward with trying it with a client…I just need technology so I can try it.” (Clinician 4) “Because it’s not always a set place. But I’ve got to just make sure I get into a quiet room, clean background, and that I’m not distracted.” (Clinician 1) “I like some of the protocols they had for emergencies. Make sure you know where the house is, address, location, what room the kid’s in, that a parent is there, which is smart if you’re doing exclusive telehealth, and dealing with someone in trauma. That just seems to be a very smart protocol that I wouldn’t necessarily have thought of.” (Clinician 1) “It was just so helpful, especially because I provide a lot of services in Spanish…they even provided a folder with resources in Spanish language…that was incredibly helpful for me and the other bilingual clinician…we didn’t have that much in our agency for Spanish only.” (Clinician 2)
Leadership	For implementation of TF-CBT telehealth program: leader commitment, involvement, and accountability	“I did talk with her…and she thought it would be a great idea to do [the training], since it is something that we’re looking to build and increase, just so that we can see more of the clients that we’re missing because of not being able to see everybody in office.” (Clinician 3) “Oh, absolutely! The director…is very supportive. She’s from a clinical background and very supportive of any kind of mental health that needs to be done. And the executive director was a therapist when the agency started, she’s very supportive as well. And the clinical director is also supportive. And so we have a lot of people in upper management that are supportive, and then lots of staff that are supported as well.” (Clinician 8) “She’s opening up more and trying to help us do [telehealth] even though she still likes in-person…She knows sometimes [in-person therapy] is not realistic.” (Clinician 7)
Clinician Attitudes	Site staff (clinicians) readiness to participate in TF-CBT telehealth training and attitudes, knowledge of TF-CBT telehealth program; competence and beliefs in capabilities to achieve implementation goals	“…that engagement portion, and then feeling a part of it, I find that it’s really important to be able to offer them something very similar to what they would have got if they were in person.” (Clinician 4) “It’s like a definite backup option, so there’s no reason to really not do therapy.” (Clinician 8) “I think there’s a time and a place for it. My preference is in person, just being able to see and be with them in person….and with our funding and so forth, preferred to have in person, and just with reimbursements. (Clinician 1) “I think there’s willingness, but I think there’s also a little bit of hesitation on the way that the client is going to respond. I think most of my coworkers prefer the face to face option.” (Clinician 2) “At first, it [TF-CBT telehealth] wasn’t something they were even looking at. And then I went through this training…then they became more open to the idea of offering telehealth, because they had wanted everything in person…. having those resources was very eye-opening for them, and opened the door to the idea of offering telehealth” (Clinician 4)
Child/Family Involvement	Degree to which family members and children are supportive and involved in TF-CBT telehealth program; includes involvement barriers and strategies to make program work more smoothly/be more feasible for families	“The fact that they show up, which means that there’s some parental support, to making sure that they have the equipment ready, and that they’re on time, and that it’s a priority.” (Clinician 5) “So [parent involvement] seems good, for the most part. The follow-up sometimes is via text or something else with them after, just kind of summing it up. It doesn’t mean you can always get the parents after the call, or even at the beginning of the call. You know, it just, the phone passes off really quickly…trying to incorporate and keep parents involved is important, and that’s a little bit of a struggle.” (Clinician 1) “I have found that if I have the proper resources with activities for them, then they’re more engaged, if it involves games, if it involves sort of virtual activities, where they can participate. Then they get to be more involved.” (Clinician 2) “…as far as my teenagers, and even fifth graders, some of them, it is really easy to do [telehealth] with them. The attention span was longer…” (Clinician 7) “I wish I could have a little bit more involvement, but this is the nature of the beast anyway, because even with in-person, trying to get them the buy-in for them to participate, that can be a struggle. I guess it doesn’t make really a difference if it was telehealth or not, whether or not they choose or how much they want to participate, the parent or guardian.” (Clinician 7) “…explaining the model, and getting some buy-in from the parents, and helping them understand what the model is going to look like, and that we’re going to do these things over telehealth…” (Clinician 5) “…and just kind of selling it. A lot of parents…when they hear I have to be home when they’re doing the service or because it’s …in the house. But then someone has to be there, or I have to be there, then it’s not as inviting I think sometimes. But I think selling it more might help.” (Clinician 6) “I don’t think that children under the age of 8 or 9 or 10 should really be on telehealth because they have a hard time focusing. A lot of times parents will leave them alone in the room with their device…And their ability to participate on telehealth is really important. So there are kiddos who experience trauma but also experience ADHD, probably not going to recommend that they do telehealth because they’re not going to focus. It’s hard enough to work with them in a room.” (Clinician 8)
Child/Family Acceptability	Perceived child/family value about health and well-being; degree to which program is satisfactory to children/families in participating sites; also includes strategies to make program more acceptable for families	“I think they appreciate it, because it does allow us to see kids and not keep them on a waitlist because of not having those evening [appointments] open. We’re able to get them in much sooner by offering telehealth option…with the cost of gas it’s much more time and cost-saving as well, because they’re not having to drive an hour to come in for a session and then drive an hour home. So it’s saving them that way.” (Clinician 3) “The kids seem to like it… 50 min is long to keep their attention. Sometimes you get that personal feel with the kids. They’ll show you their room or their house, maybe some artwork that they’re working on…So that’s kind of fun. Sometimes we do origami…like a simple little folding kind of pattern, making a little boat or something that has a little story to it. So it seems like they like that.” (Clinician 1) “And then also the kids can be much more at ease because they’re in a place that they know and they’re already comfortable with.” (Clinician 3) “And I think for the adolescent group, some of them actually prefer to meet via telehealth more than in person, because they’re in the comfort of their own home, or they can be in their space, in their room…they just enjoy that comfort of home. And I have seen them for a long time.” (Clinician 2) “My teenagers love it…look forward to it. I don’t have a lot of absenteeism with them…” (Clinician 7)

**Table 2 healthcare-12-02110-t002:** Exemplary quotations: implementation outcomes and strategies for TF-CBT via telehealth.

**Implementation Outcomes**		
**Code**	**Definition**	**Quotations**
Adoption	Description of ‘initial decision’ steps within organization to take on TF-CBT telehealth training and program	“It seems like most of my clients, I’ve met with at least one time, in person visit, is kind of our office protocol. Try to meet with them, do the initial assessment, and then evaluate where their needs are. And thinking of the process, it seemed like it went well initially, and that [telehealth] would be convenient for them, it would be a good thing for them, trying to see if it was a good fit.” (Clinician 1) “So being able to offer telehealth was a big draw for us. [A co-therapist] has been doing [telehealth] because of COVID, without having training and kind of muddling along. She’s going to take the training soon, just because I’ve sung the praises of how much I’ve learned from it and how much I’ve enjoyed it.” (Clinician 3)
Trauma-Based Care Penetration	Delivery and completion of telehealth program in site setting and integrated within existing care (completion of telehealth visit, follow-up visits)	“For me personally, when I’m doing intakes for my clients, I look at age, diagnosis, and at their ability to tolerate being on telehealth.” (Clinician 8) “Every referral that comes in gets asked if they would like to consider telehealth.” (Clinician 3) “I don’t do as much as I used to. I used to do a lot…right after we got out of quarantine and people started returning to school, and a lot of things were done virtually. Now, I could say out of five of my clients, I only do one telehealth, and that’s because she’s in a different locality. She’s 45 min away. So that’s the only way we’re able to do sessions.” (Clinician 7)
Feasibility/ Acceptability	Clinician and site team perceptions of telehealth training and program, including extent to which training and program can be successfully conducted in site and degree to which program is agreeable or satisfactory	“I feel in some cases it’s been very successful, and it accesses those that have a difficult time coming, or distance, possible language barriers. I’m also fluent in Spanish, and so I provide therapy in Spanish. We live in a rural area, so sometimes it was very effective when we were able to do our calls on a platform…and have a videoconferencing. It seemed very easy for the client to do.” (Clinician 1) “I think it really has exceeded my expectations, because of the access to resources that I didn’t have before. I think that has been key.” (Clinician 2) “[Consultation calls] were helpful, because I was able to connect with other clinicians, and in a sense offer me some normalizing, that a lot of my challenges were also faced by other clinicians. And then the sharing, again, sharing resources, sharing ways of doing things among each other, and obtaining guidance from the trainers on how to implement certain services.” (Clinician 2) “I thought the training was fantastic. I learned a lot of things that I hadn’t thought about in starting telehealth, in general, but with TF-CBT. And being so new with TF-CBT…[it was] a priority for me to be able to see TF-CBT at work in the telehealth realm. Because, doing it in person and doing it through a computer, it is different. So that really appealed to me because we are in such a rural area, so that really was a draw for me to be able to take [training]…and grow my knowledge of TF-CBT in that aspect.” (Clinician 3)
Patient Effectiveness Outcomes	Change in patient outcomes resulting from program	“But I think that it’s doable and effective…We met our goals. They’re doing better. They’ve been able to stay at school, or their emotional regulation skills are better…parents report their kids were doing better, so I think that that reflects to the model and delivery of telehealth.” (Clinician 5) “…the treatment that we provide [via telehealth] has been just as successful as if it was face to face.” (Clinician 2)
Sustainability	Extent to which newly implemented practice will be maintained within site’s ongoing operations	“As we’re going two years into it now…our director has been more open to getting us tools we need to be able to do [telehealth] properly. Just recently she spent TF funding to purchase us headsets, really nice headsets for our sessions. She upgraded some of our laptops and computers so that we could do our sessions. Some of them didn’t have the proper camera or they were outdated…then upgraded our systems, like our internet, so it could support it. So she’s opening up more and trying to help us, do [telehealth], even though she still likes the in-person.” (Clinician 7)
**Implementation Strategies**		
Education	MUSC training and education of site stakeholders about telehealth	“I would like to see a little bit more of…how different it is to provide services to communities, specific communities…not only Spanish-speaking, but also children with developmental disabilities, or children with ADHD or other conditions. Those kids won’t stay on a video call the whole session…” (Clinician 2) “The only other thing that, at least for me, that would be helpful, is a lot of the resources in the [online folder] seem to be geared towards the younger kids. Maybe…having some things that are more for the teenagers.” (Clinician 3) “…I feel at some point it would be nice to retake the telehealth training to get updated on new technology that might be helpful, or a new way of presenting something.” (Clinician 4) “Every now and then, to maybe have a check-in consultation call to just see what new problems have arisen a year later. Like what are we seeing now that things have changed post-COVID, or what new things and resources are out there that we could now get from our trainers? Because I’m sure there’s new ones.” (Clinician 5)
Technical assistance	MUSC helping with technology and other set up and implementation issues	“We do have IT support that are off site. But as far as during sessions, if there’s a problem we usually troubleshoot it ourselves. And then if we can’t fix it then we would get our IT person involved.” (Clinician 3) “We do have an IT person in the agency as a whole, but it’s not to help us with telehealth services. It’s more for general IT needs…But I think the clinicians do their own. When we got the iPads, we set up our own iPads that we were going to give to our clients.” (Clinician 2) “I think just a little more help with the technology side of using the resources, sharing screens, some of that. Not everyone is familiar with that, including myself….But there are great resources they have. I’ve seen them. But just knowing how to use them a little bit more.” (Clinician 1)

## Data Availability

The interview data are unavailable due to privacy concerns.

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
