# Peer review of "Telehealth Outreach Program for Child Traumatic Stress: Strategies for Long-Term Sustainability"

_healthcare, 2024, doi:10.3390/healthcare12212110_

Round 1

Reviewer 1 Report

Comments and Suggestions for Authors

The manuscript titled "Telehealth Outreach Program for Child Traumatic Stress: Strategies for Long-Term Sustainability" presents an original and timely study addressing the disparities in access to trauma-focused mental health care for children, particularly among racial and ethnic minority youth and those in rural areas. The paper explores the implementation of Trauma-Focused Cognitive Behavioral Therapy (TF-CBT) via telehealth and identifies key barriers and facilitators to sustaining this intervention post-training. Using a mixed-methods approach, the authors collected data from clinical sites, highlighting varied program adoption, leadership involvement, community needs, and logistical challenges. The study’s conclusions focus on strategies for improving the long-term sustainability of telehealth-delivered TF-CBT, with the goal of increasing access to quality care for underserved populations.

While the study addresses an important topic, there are areas that require additional clarification and improvement:

  • Participants (Section 2.3): More detailed demographic information is needed, such as the participants’ average age, gender, specialization, years of service, country of origin, etc.. It would also be important to note if a participation consent form was signed in accordance with ethical guidelines.

  • Data (Section 2.4): The survey used in this study needs to be described in more detail, as readers may want to understand its structure and content.

  • Data Analysis (Section 2.5): This section requires a clearer explanation of the step-by-step methodology used to analyze the data. The clustering method mentioned is unclear—how were these clusters identified and organized? Have you followed Grounded theory or other theories? Additionally, a brief explanation of NVIVO software should be provided for readers unfamiliar with it. In Table 1 and 2, quotations should be accompanied by participant codes for clarity.

  • Results (Section 3.1): It is unclear why only 15 surveys were collected from 102 invitations. The authors should clarify the low response rate and specify strategies used to mitigate this issue.

  • Tables 1 and 2: Both tables need brief introductions explaining what the readers will find in them, as the current structure is confusing.

  • Limitations and Future Directions: The manuscript would benefit from a section discussing the study's limitations and suggestions for future research.

  • Sample Size: The number of participants is too low to draw significant conclusions, even for a pilot study, especially given the vast number of mental health providers in the regions studied. I recommend conducting additional interviews and collecting more surveys to increase the sample size and enhance the study’s validity. 

In summary, while the paper is original and well-written, these adjustments are necessary to improve clarity and strengthen the overall contribution of the study.

Author Response

Please see attached for response to reviewer 1 feedback.

Reviewer 2 Report

Comments and Suggestions for Authors

The current  study examines the implementation of Trauma-Focused Cognitive Behavioral Therapy (TF-CBT) via telehealth in underserved populations. It focused on barriers and facilitators to its delivery. It emphasizes the role of leadership support, resource availability, and technology as key facilitators, while challenges such as limited patient technology access and engagement issues were identified. With its limitations, it is a significant original study that could contribute to current literature.

1.       Authors should more clearly define the gap this study fills and what it adds to current literature.

2.       Provide more details about EPIS model and the adaptation of the EPIS model to the TF-CBT setting. What specific aspects were adapted, and how did this adaptation influence the study design?

3.       Clarify the recruitment strategy. Specify how the mental providers were identified or selected. What was the target population? Were there inclusion/exclusion criteria? More details needed on the recruitment process. Provide rationale why only participants who completed the training in the prior two years were invited to participate.

4.       The development of the self-report survey should be explained detail. How were the survey questions were created? What was pilot testing findings? Was it pre-tested, or based on validated instruments? Are there any reliability and validity findings?

5.       Implementation of survey on online should be detailed. What was its duration? How many screens was it composed of? How did you handle missing responses? What did you do to maintain confidentiality?

6.       The same also applies to description of data collection for interviews. What was the structure of the interviews? What is meant by the standardized guide? Detail the description of the interview format, probes, and structure. Also identify the female interviewers, their competencies and training.

7.       Data analysis section should be expanded. Explain how codes were developed and refined, how was process of coding, what was the criteria for updating the codebook, and how saturation was determined

8.       The discussion mainly reiterates information already presented in the introduction and methods. The discussion should focus more on interpreting and synthesizing the study's findings. Please revise relevant parts.

Overall the study would be a beneficial one to the readers after these  revisions.

Author Response

Please see attached for response to reviewer 2 feedback.

Round 2

Reviewer 1 Report

Comments and Suggestions for Authors

The revised manuscript has addressed all the suggestions and comments raised in the previous review, and no issues were detected upon further evaluation. The authors have satisfactorily incorporated the necessary changes, ensuring that the manuscript now meets the required standards. Therefore, I find the manuscript suitable for publication in its current form.